Original research

# Feeding practices and growth patterns of moderately low birthweight infants in resource-limited settings: results from a multisite, longitudinal observational study

Linda Vesel ,[1] Roopa M Bellad,[2] Karim Manji ,[3] Friday Saidi ,[4]
Esther Velasquez,[5] Christopher R Sudfeld ,[6] Katharine Miller,[1] Mohamed Bakari,[3]
Kristina Lugangira,[3] Rodrick Kisenge,[3] Nahya Salim,[3] Sarah Somji,[3]
Irving Hoffman,[7] Kingsly Msimuko,[4] Tisungane Mvalo,[4,8] Fadire Nyirenda,[4]
Melda Phiri,[4] Leena Das,[9] Sangappa Dhaded ,[2] Shivaprasad S Goudar,[2]
Veena Herekar,[2] Yogesh Kumar,[2] M B Koujalagi,[10] Gowdar Guruprasad,[10]
Sanghamitra Panda,[11] Latha G Shamanur,[12] Manjunath Somannavar,[2]
Sunil S Vernekar,[2] Sujata Misra,[9] Linda Adair,[13] Griffith Bell,[1] Bethany A Caruso,[14]
Christopher Duggan,[15] Katelyn Fleming,[1] Kiersten Israel-Ballard,[16] Eliza Fishman,[1]
Anne C C Lee ,[17] Stuart Lipsitz,[1] Kimberly L Mansen,[16] Stephanie L Martin,[13]
Rana R Mokhtar,[1] Krysten North,[17,18] Arthur Pote,[1] Lauren Spigel,[1]
Danielle E Tuller,[1] Melissa Young,[14] Katherine E A Semrau ,[1] The LIFE study
team

For numbered affiliations see end of article.

**Correspondence to**
Dr Linda Vesel;
lvesel@ariadnelabs.org

## ABSTRACT

**Objectives** To describe the feeding profile of low birthweight (LBW) infants in the first half of infancy; and to examine growth patterns and early risk factors of poor 6-month growth outcomes.

**Design** Prospective observational cohort study.

**Setting and participants** Stable, moderately LBW (1.50 to <2.50 kg) infants were enrolled at birth from 12 secondary/tertiary facilities in India, Malawi and Tanzania and visited nine times over 6 months.

**Variables of interest** Key variables of interest included birth weight, LBW type (combination of preterm/term status and size-for-gestational age at birth), lactation practices and support, feeding profile, birthweight regain by 2 weeks of age and poor 6-month growth outcomes.

**Results** Between 13 September 2019 and 27 January 2021, 1114 infants were enrolled, comprising 4 LBW types. 363 (37.3%) infants initiated early breast feeding and 425 (43.8%) were exclusively breastfed to 6 months. 231 (22.3%) did not regain birthweight by 2 weeks; at 6 months, 280 (32.6%) were stunted, 222 (25.8%) underweight and 88 (10.2%) wasted. Preterm-small-for-gestational age (SGA) infants had 1.89 (95% CI 1.37 to 2.62) and 2.32 (95% CI 1.48 to 3.62) times greater risks of being stunted and underweight at 6 months compared with preterm-appropriate-for-gestational age (AGA) infants. Term-SGA infants had 2.33 (95% CI 1.77 to 3.08), 2.89 (95% CI 1.97 to 4.24) and 1.99 (95% CI 1.13 to 3.51) times higher risks of being stunted, underweight and wasted compared with preterm-AGA infants. Those not regaining their birthweight by 2 weeks had 1.51 (95% CI 1.23 to 1.85) and 1.55 (95% CI 1.21 to 1.99) times greater risks of being stunted and underweight compared with infants regaining.

**Conclusion** LBW type, particularly SGA regardless of preterm or term status, and lack of birthweight regain by 2 weeks are important risk identification parameters. Early interventions are needed that include optimal feeding support, action-oriented growth monitoring and understanding of the needs and growth patterns of SGA infants to enable appropriate weight gain and proactive management of vulnerable infants.

**Trial registration number** NCT04002908.

## STRENGTHS AND LIMITATIONS OF THIS STUDY

⇒ This study collects 6-month longitudinal data on feeding and growth patterns among moderately low birthweight infants (1.50 to <2.50 kg) in resource-limited settings.

⇒ Linear mixed-effects models were used to assess differences in growth by low birthweight type.

⇒ A quasi-likelihood estimation approach specifying a generalised linear model with a log link, a Poisson distribution and robust SEs was used to estimate relative risks.

⇒ When determining generalisability of the findings, it should be noted that our cohort had ready access to secondary/tertiary health facilities; and the majority of preterm infants were late-preterm, thus influencing feeding patterns/abilities.

## INTRODUCTION

Low birthweight (LBW; <2.50 kg) infants account for nearly 15% of births, yet make up 80% of neonatal deaths.[1] Three-quarters of the world's LBW infants reside in sub-Saharan Africa and South Asia.[1] LBW infants include those born premature (<37 weeks gestation) and/or small-for-gestational age (SGA; <10th percentile of weight for gestational age). The majority of LBW infants are moderately LBW (1.50 to <2.50 kg).[1 2] Furthermore, LBW infants are at increased risk for morbidity, growth deficits, chronic conditions and neurodevelopmental delays compared with those with birthweight ≥2.50 kg.[1 3] LBW infants are also known to experience delays in feeding initiation, feeding difficulties and barriers to exclusive breast feeding (EBF).[4 5]

There is little evidence on the standard of care, feeding practices, growth patterns and associated health outcomes among LBW infants in low-income and middle-income countries (LMICs). Available literature is concentrated on very LBW (<1.50 kg) or preterm infants and in high-income settings.[6] Three-quarters of the recommendations in the 2011 WHO LBW infant feeding guidelines are based on low/very low quality evidence (more recent recommendations were released in 2022 after this study was completed).[7] This gap in knowledge makes it difficult to design and test rigorous and sustainable interventions to prevent and manage growth faltering among LBW infants.

There is renewed interest and investment in efforts to improve quality of care provision, and to prevent and manage poor outcomes among LBW infants.[8 9] In addition to Sustainable Development Goal 3.2, which aims to reduce preventable child deaths by 2030, key stakeholders put out an urgent call to action in 2017 seeking more evidence on the feeding and care of sick and vulnerable newborns.[10 11] The MAMI Care Pathway group has focused on highlighting the need for actionable evidence to improve quality of care provision and to prevent and manage poor outcomes in infants in the first 6 months of life, including LBW infants and their mothers.[12] As the COVID-19 pandemic has caused major disruptions in care and stretched health systems globally, a focus on the already vulnerable is even more critical.[9] The LBW Infant Feeding Exploration (LIFE) study aimed to understand feeding practices, growth patterns and other health outcomes among moderately LBW infants in India, Malawi and Tanzania.[13] This paper addresses two objectives: (1) to describe the feeding profile of LBW infants in the first half of infancy and (2) to examine growth patterns and early risk factors of poor growth outcomes at 6 months.

## METHODS
### Study design and participants

The LIFE study is a formative, multisite, prospective, observational cohort study using a convergent parallel, mixed-methods design to establish foundational knowledge needed to design feeding interventions for nutritionally at-risk LBW infants.[13] To achieve the overall study aim, multiple data collection activities were employed including a retrospective chart review, 1-year prospective observational cohort, birth to discharge in-facility observational cohort, facility needs assessment, in-depth interviews and focus group discussions to examine feeding patterns, key health outcomes, health system inputs for the care and feeding of LBW infants, and the barriers and facilitators of infant feeding practices from the perspective of diverse respondents. In this paper, we present results from the first 6 months of the prospective cohort study, conducted between 13 September 2019 and 27 January 2021. Data were collected for mother–infant dyads recruited from 12 health facilities in 4 sites in India (5 in Belgaum and Davangere, Karnataka State; 2 in Cuttack, Odisha state), Malawi (2 in Lilongwe) and Tanzania (3 in Dar-es-Salaam). All facilities in India-Odisha, Malawi and Tanzania were public and secondary/tertiary; in India-Karnataka, all hospitals were tertiary with a mix of public and private.[13]

Inclusion criteria were birthweight from 1.50 to <2.50 kg and residence within 50 km of the enrolment facility. Infants were excluded if born with congenital abnormalities impacting feeding, had a twin who died prior to screening, had mothers younger than 16–18 years of age (dependent on site) or had mothers who died after screening but prior to consent for data collection. Further exclusions from analysis included infant death <72 hours of birth or administrative withdrawal from the study.[13]

### Procedures

Mother–infant dyads were screened for eligibility at study facilities within 72 hours of birth using a checklist based on chart data and maternal interviews. If the dyad met inclusion criteria, they were enrolled, and a baseline assessment was conducted via chart checks, maternal interviews and examinations. The assessment captured demographic characteristics, pregnancy history, feeding since birth, anthropometrics and Dubowitz assessment of gestational age (GA). Dyads were visited at nine time points (baseline/0–72 hours and 1, 2, 4, 6, 10, 14, 18 and 26 weeks of age) in the facility or community via maternal interviews, observations and examinations. Detailed descriptions of follow-up assessments have been published in the protocol.[13]

Prior to data collection, site investigators conducted training that covered standard operating procedures, surveys, anthropometric measurement and calibration of equipment. Refresher training and data quality assurance were conducted throughout to ensure consistency and accuracy of data collection. Surveys were translated into local languages. Infant anthropometric measurements included weight, length, head circumference and mid-upper arm circumference. In this paper, we focus on reporting findings related to weight and length, common indicators of attained size; results related to other measures will be shared in a separate publication.

Anthropometrics were collected in triplicate at each of the nine study visits using standardised equipment (SECA 334 mobile digital baby scale, SECA 417 infantometer and Shorr MUAC tapes—SKU: WM-S Tape) and calibration protocols.[13] Weight was also measured at birth prior to the study baseline assessment; birth length was not collected. We calculated the median of the three measurements, retained those that were within ±10 g of the median for weight and ±0.5 cm of the median for the other measurements, and calculated the mean of the remaining measurements.

We employed a safety net standard operating procedure, including the completion of a safety net module by data collectors at all study visits. The indicators that we assessed included: malnutrition/severe growth faltering (ie, failure to regain birthweight by 4 weeks, weight-for-length (WLZ) z-score <−3, oedema in both feet and visible wasting), danger signs or severe illness among the mother or infant. When study staff encountered a mother–infant dyad with any indication, a subsequent referral for advanced care was completed.

## Definition of key variables
### Birthweight
Birthweight, used for eligibility assessment and all relevant analyses, was measured by facility staff using local equipment prior to baseline assessment and recorded from patient charts. Birthweight was adjusted based on the time between birth and weighing using an established algorithm to account for weight loss (ie, three infants had birthweight of 1.46–1.49 kg due to weighing at 43–54 hours after birth).[14] Birthweight regain by 2 weeks, a dichotomous indicator, was defined as an infant attaining or exceeding birthweight by the 2-week visit.

### Age
Chronological age is the time in weeks since birth. Postmenstrual age (PMA) is chronological age plus GA at birth in weeks. Unlike chronological age, PMA accounts for the maturity of an infant (ie, GA corrected) and allows for consideration of biological differences given GA at birth.

### LBW type
Infants were stratified into four LBW types at birth based on GA (<37 weeks or ≥37 weeks) and size-for-GA (SGA; appropriate-for-GA (AGA), 10–90th percentile of weight for GA; and large-for-GA (LGA), >90th percentile): (1) preterm-SGA, (2) preterm-AGA, (3) preterm-LGA and (4) term-SGA.[15 16] Infants with missing (n=1) or implausible (GA <24 weeks; n=1) GAs were excluded from the analysis. GA determination was prioritised by best obstetric estimate combining first/second trimester ultrasound and last menstrual period (LMP) recorded in the chart. When not available, we used LMP based on maternal reports, followed by GA in chart no matter the source, and finally Dubowitz examination at birth.

### Feeding
Feeding profile was based on 7-day recall of feeding patterns at each visit: human milk only, mixed milk feeding (human milk and another liquid/solid food, including formula, animal milk or water), no human milk and not yet fed.[17] We defined EBF to 6 months as having been fed only human milk (direct from the breast or expressed) at each of the eight visits prior to the 6-month visit, allowing for provision of oral rehydration solution, drops and syrups (vitamins, minerals and medicines).[17] Lactation support at the baseline visit was defined as the receipt of any type of support (verbal or physical) in an individual/group format based on maternal report.

### Growth
Six-month stunted, underweight and wasted were defined by <−2 z-scores for length-for-age (LAZ), weight-for-age (WAZ) and WLZ, respectively. WHO growth standards were used for term infants and INTERGROWTH-21st standards were used for preterm infants to calculate z-scores and size-for-GA; at 6 months, we used WHO standards corrected for GA for preterm infants.[15 16]

## Statistical analysis
We calculated the sample size for precision of estimates based on the percent of LBW infants whose mean LAZ at 6 months was <−2 z-scores; with 300 dyads per site, we had precision of at least ±3.6% for a true proportion of 10% of infants.[13] We first conducted descriptive analyses of maternal and infant characteristics, LBW type, feeding patterns and growth indicators using means, SD or frequencies and percentages. We next assessed the relationship between LBW type and postnatal growth patterns from birth to 6 months using two types of crude and multivariable linear mixed-effects models: (1) changes in mean WAZ, LAZ and WLZ with an interaction term between LBW type and chronological age in weeks based on study visit and (2) changes in mean weight and length with an interaction term between LBW type and PMA in weeks grouped into 14 categories. An interaction term between LBW type and visit week was used to assess the statistical significance of differences in growth over time by LBW type. Finally, we evaluated two individual models with 6-month growth as the outcome, and LBW type and lack of birthweight regain by 2 weeks as the respective exposures. For models with binomial outcomes, we estimated relative risks using a quasi-likelihood estimation approach specifying a generalised linear model with a log link, a Poisson distribution and robust SEs.[18 19] For models with continuous outcomes, we estimated mean differences using linear regression. All models used a compound symmetry working correlation matrix to account for correlations for multiple births; were clustered by mother to account for twins; and adjusted for potential confounders, including maternal education, maternal age, parity, wealth quintile, place of residence, infant sex, birthcount, LBW type (when not an exposure) and study site. We pooled data across sites and adjusted

for site as a covariate rather than stratifying to retain study power and make general conclusions about the associations in LMIC. Analyses were conducted by using StataBE V.17, R V.4.1.2 and SAS V.9.

The study is registered with ClinicalTrials.gov (NCT04002908) and CTRI/2019/02/017475 (Clinical Trial Registry of India - http://ctri.nic.in).[13]

## Patient and public involvement

Study tools were piloted with local stakeholders (eg, mothers of LBW infants, community members and clinicians) to ensure that research questions were culturally appropriate, understandable and relevant to the study population. The study team involved personnel familiar with each of the respective settings and populations.

## RESULTS
## Characteristics of LBW infants

After screening 1982 mothers and 2152 infants for eligibility, 1114 infants and their mothers (n=1070) were enrolled and analysed (figure 1).[20] Six-month follow-up was completed for 985 (88.4%) infants and 940 (87.9%) mothers; 972 (87.3%) infants had eight or more visits (of nine) completed. While data collection took place between 13 September 2019 and 27 January 2021, there were site differences in start/end dates and pace of recruitment due to the timing of ethics approvals and degree of disruptions from the COVID-19 pandemic. When visits could not be safely conducted in-person or participants did not consent to home visits, data collection was completed via phone. In total, 127 (12.9%) infants

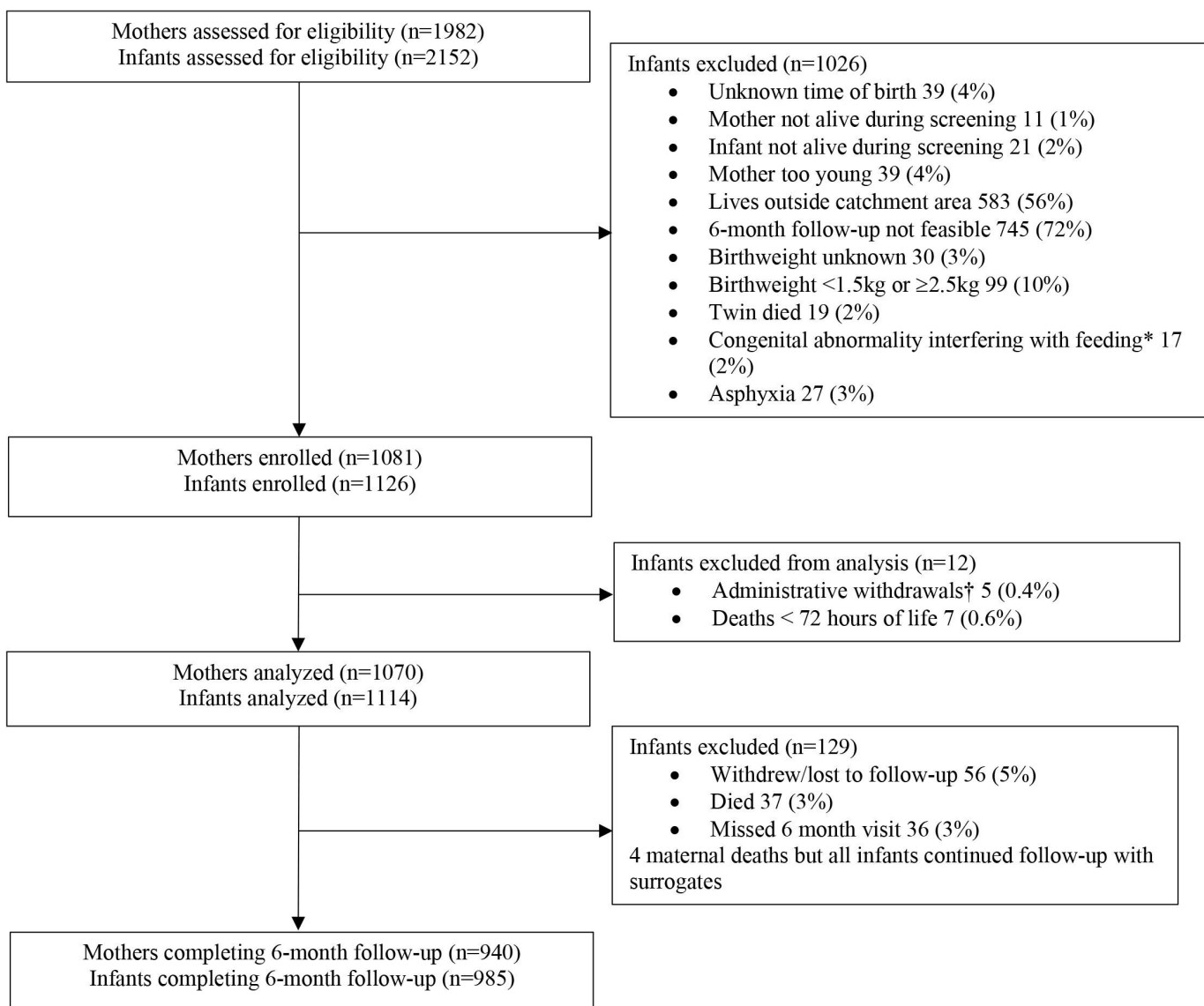

**Figure 1** Flow chart. Exclusion criteria for enrolment are not mutually exclusive. *Congenital abnormalities interfering with feeding included cleft palate (n=4), hydrocephalus (n=4), gastrointestinal anomalies (n=6), neural tube defect (n=3), congenital cardiac defect (n=4), trisomy (n=1) and toxoplasmosis, other agents, rubella, cytomegalovirus and herpes simplex (n=3). †Administrative withdrawals refer to infants who were withdrawn from the study by investigators in the first 72 hours because they were dually enrolled in a different research study at the same health facility.

**Table 1** Baseline characteristics of moderately low birthweight infants and their mothers in four sites

| Maternal characteristics | | India Karnataka N=300 | India Odisha N=197 | Malawi N=273 | Tanzania N=300 | Total N=1070 |
|---|---|---|---|---|---|---|
| Maternal age (in years), mean (SD) | | 24.4 (3.9) | 25.0 (4.3) | 25.2 (6.1) | 26.7 (5.8) | 25.4 (5.2) |
| Maternal education, n (%) | Primary or less | 67 (22.3) | 105 (53.3) | 155 (56.8) | 174 (58.0) | 501 (46.8) |
| | Secondary or more | 233 (77.7) | 92 (46.7) | 118 (43.2) | 126 (42.0) | 569 (53.2) |
| Marital status, n (%) | Married | 300 (100) | 197 (100) | 254 (93.0) | 264 (88.0) | 1015 (94.9) |
| Residence, n (%) | Urban | 173 (57.7) | 135 (68.5) | 211 (77.3) | 300 (100) | 819 (76.5) |
| | Rural | 126 (42.0) | 61 (31.0) | 62 (22.7) | 0 (0) | 249 (23.3) |
| | Missing | 1 (0.3) | 1 (0.5) | 0 (0) | 0 (0) | 2 (0.2) |
| Mother's parity, n (%) | One birth | 169 (56.3) | 119 (60.4) | 104 (38.1) | 123 (41.0) | 515 (48.1) |
| | Two births | 88 (29.3) | 61 (31.0) | 59 (21.6) | 80 (26.7) | 288 (26.9) |
| | 3+ births | 43 (14.3) | 17 (8.6) | 110 (40.3) | 97 (32.3) | 267 (25.0) |
| Any antenatal care visit attendance, n (%) | Yes | 299 (99.7) | 184 (93.4) | 270 (98.9) | 299 (99.7) | 1052 (98.3) |
| No infants in delivery, n (%) | Singleton | 287 (95.7) | 197 (100) | 222 (81.3) | 250 (83.3) | 956 (89.4) |
| | Twins | 13 (4.3) | 0 (0) | 51 (18.7) | 50 (16.7) | 114 (10.7) |
| Maternal positive HIV status, n (%) | Yes | 0 (0) | 0 (0) | 33 (12.1) | 16 (5.3) | 49 (4.6) |
| | No information available | 1 (0.3) | 0 (0) | 16 (5.9) | 1 (0.3) | 18 (1.7) |
| Wealth quintile, n (%) | Bottom 20% | 67 (22.3) | 37 (18.8) | 80 (29.3) | 72 (24.0) | 256 (23.9) |
| | 20%–40% | 51 (17.0) | 39 (19.8) | 34 (12.5) | 45 (15.0) | 169 (15.8) |
| | 40%–60% | 65 (21.7) | 41 (20.8) | 27 (9.9) | 62 (20.7) | 195 (18.2) |
| | 60%–80% | 53 (17.7) | 32 (16.2) | 49 (17.9) | 61 (20.3) | 195 (18.2) |
| | Top 20% | 55 (18.3) | 36 (18.3) | 48 (17.6) | 55 (18.3) | 194 (18.1) |
| | Missing | 9 (3.0) | 12 (6.1) | 35 (12.8) | 5 (1.7) | 61 (5.7) |
| Infant characteristics | | N=309 | N=197 | N=300 | N=308 | N=1114 |
| Infant sex, n (%) | Female | 166 (53.7) | 115 (58.4) | 156 (52.0) | 171 (55.5) | 608 (54.6) |
| Sibling at birth (of those enrolled), n (%)* | Singleton | 291 (94.2) | 197 (100) | 246 (82.0) | 292 (94.8) | 1026 (92.1) |
| | Twin | 18 (5.8) | 0 (0) | 54 (18.0) | 16 (5.2) | 88 (7.9) |
| Place of birth, n (%) | Study facility | 299 (96.8) | 197 (100) | 262 (87.3) | 279 (90.6) | 1037 (93.1) |
| | Outside study facility | 10 (3.2) | 0 (0.0) | 38 (12.7) | 29 (9.4) | 77 (6.9) |
| Delivery mode, n (%) | Vaginal | 160 (51.8) | 137 (69.5) | 269 (89.7) | 198 (64) | 764 (68.6) |
| | Caesarean section | 149 (48.2) | 60 (35.9) | 31 (10.3) | 110 (35.7) | 350 (31.4) |
| Gestational age (in weeks), mean (SD) | | 37.3 (2.2) | 38.0 (2.1) | 35.9 (2.6) | 35.7 (3.0) | 36.6 (2.7) |
| Birthweight (in g), mean (SD) | | 2151 (248) | 2195 (197) | 2091 (245) | 2041 (252) | 2112 (246) |
| Birthweight band (in g), n (%)† | 1500–1749 | 28 (9.1) | 6 (3.1) | 31 (10.3) | 39 (12.7) | 104 (9.3) |
| | 1750–1999 | 41 (13.3) | 20 (10.2) | 49 (16.3) | 74 (24.0) | 184 (16.5) |
| | 2000–2499 | 240 (77.7) | 171 (86.8) | 220 (73.3) | 195 (63.3) | 826 (74.2) |
| LBW type, n (%) | Preterm SGA | 42 (13.6) | 15 (7.6) | 38 (12.7) | 56 (18.2) | 151 (13.6) |
| | Preterm AGA | 62 (20.1) | 28 (14.2) | 107 (35.7) | 130 (42.2) | 327 (29.4) |
| | Preterm LGA | 2 (0.7) | 1 (0.5) | 16 (5.3) | 18 (5.8) | 37 (3.3) |
| | Term SGA | 203 (65.7) | 153 (77.7) | 138 (46.0) | 103 (33.4) | 597 (53.7) |
| | Implausible or missing‡ | 0 (0.0) | 0 (0.0) | 1 (0.3) | 1 (0.3) | 2 (0.2) |
| Birthweight by LBW type (in g), mean (SD) | Preterm SGA | 1882 (205) | 1897 (213) | 1917 (209) | 1876 (185) | 1890 (198) |
| | Preterm AGA | 2073 (266) | 2157 (206) | 2084 (235) | 2077 (254) | 2085 (247) |
| | Preterm LGA | 2100 (424) | 2000§ | 2092 (237) | 1978 (317) | 2034 (281) |
| | Term SGA | 2231 (200) | 2233 (167) | 2145 (243) | 2093 (232) | 2187 (216) |

Continued

**Table 1**  Continued

| Maternal characteristics | | India Karnataka  N=300 | India Odisha  N=197 | Malawi  N=273 | Tanzania  N=300 | Total  N=1070 |
|---|---|---|---|---|---|---|
| Infant length of stay (days), mean (SD)¶ | Available data | 4.9 (1.7) | 4.2 (2.1) | 2.7 (1.4) | 4.3 (1.8) | 4.0 (1.9) |
|  | Missing data | 29 (9.4%) | 17 (8.6%) | 53 (17.7%) | 48 (15.6%) | 147 (13.2%) |

*Number of infants enrolled with their sibling versus without.
†Based on International Classification of Diseases.
‡This includes one infant with an implausible gestational age (<24 weeks) for which size-for-gestational age could not be calculated using the INTERGROWTH-21st newborn size at birth calculator and one infant with a missing gestational age.
§Only one infant.
¶Assessed among those who were discharged by the week one visit (7–10 days after birth).
AGA, appropriate-for-gestational age; LBW, low birthweight; LGA, large-for-gestational age; SGA, small-for-gestational age.

had a 6-month visit conducted over the phone, predominantly in Indian sites (58 (16.9%) in India-Karnataka, 68 (30.0%) in India-Odisha and 1 (0.4%) in Malawi).

The pooled cohort included 151 (13.6%) preterm-SGA, 327 (29.4%) preterm-AGA, 37 (3.3%) preterm-LGA and 597 (53.7%) term-SGA infants. African infants were predominantly preterm compared with term-SGA in India. The mean GA was 36.6 weeks (SD 2.7). The majority of preterm infants (465 (90.3%)) were late-preterm. Mean birthweight was 2112 g (246), slightly lower in African compared with Indian sites (table 1).

Infants not discharged within the first week of life were not included in this analysis, thus resulting in smaller denominators.

### Feeding of LBW infants

Overall, 363 (37.3%) infants initiated breast feeding within 1 hour of birth (299 (43.4%) delivered vaginally, 65 (22.7%) via caesarean section); more term-SGA infants initiated early breast feeding compared with the other LBW types, with large site differences (India-Karnataka 144 (52.4%), India-Odisha 144 (75.0%), Malawi 64 (24.5%) and Tanzania 12 (4.9%)) (online supplemental table S1). At baseline (26.6 hours (15.5)), 94 (8.4%) infants were not yet fed anything, fewer in India-Odisha (2 (1.0%)) and Malawi (7 (2.3%)) than in India-Karnataka (30 (9.7%)) and Tanzania (55 (17.9%)) (online supplemental tables S1 and S2). Prelacteal feeds were rare (online supplemental table S3). Human milk feeding (ie, only human milk or mixed milk feeding) was the most common (≥91.2%) feeding profile at each visit before 6 months; most fed directly from the breast (online supplemental tables S2 and S4). When expressed human milk was fed, it was expressed by hand (without a pump) and almost exclusively delivered via cup/spoon/palladai. Feeding of donor human milk was rare and only available at 1 of 12 facilities (online supplemental table S3). Mixed milk feeding increased over time (26 (2.3%) at baseline to 257 (26.5%) at week 18), most commonly observed in India-Odisha of all sites (online supplemental table S2). Not feeding human milk (ie, feeding formula, animal milk, water or any other liquid/food) was rare; feeding of formula and animal milk increased over time as did feeding of formula via a bottle. Formula was not readily available in most facilities, and only half had powdered formula in stock. Term versus preterm formula was more commonly fed and was seldom ready-made (online supplemental table S5). However, only 425 (43.8%) infants were EBF to 6 months; there were no major differences in EBF prevalence by LBW type (online supplemental table S1).

During the first 6 months, 493 (44.3%) mothers reported feeding difficulties (online supplemental table S1). Among those reporting specific difficulties (n=295), the main ones included insufficient milk (204 (69.2%)), distracted baby (55 (18.6%)), long time for milk to come in (52 (17.6%)), no milk let down (50 (17.0%)) and breast pain (50 (17.0%)). Lactation support and feeding counselling ≤72 hours was received by 855 (77.1%) mother–infant dyads, slightly more among preterm-AGA infants and their mothers than the other LBW types (online supplemental table S1). Most were given generalised advice on proper latching/positioning (753 (87.9%)); and some received physical support with positioning (556 (64.9%)), latching (396 (46.2%)), human milk expression (367 (42.8%)) or feeding with a bottle/cup/palladai (271 (31.6%)). Almost all counselled dyads (806 (94.1%)) were supported by healthcare providers (ie, doctor, nurse, midwife); family members only supported 147 (17.2%) dyads (online supplemental table S6).

### Poor growth outcomes at 6 months and early predictors

Overall, 231 (22.3%) infants did not regain their birth weight by 2 weeks, slightly more pervasive among preterm (131 (27.6%)) compared with term (100 (17.8%)) infants. Stunted was the most prevalent poor growth indicator at 6 months (280 (32.6%)), followed by underweight (222 (25.8%)) and wasted (88 (10.2%)). Stunted prevalence was higher in African sites while underweight and wasted were more common in Indian sites.

Poor 6-month growth outcomes differed by LBW type and were most prevalent among term-SGA infants (figure 2, table 2). In multivariable models, preterm-SGA infants had 1.89 (95% CI 1.37 to 2.62) and 2.32 (95% CI 1.48 to 3.62) times greater risks for 6 month stunted and underweight, respectively, compared with preterm-AGA

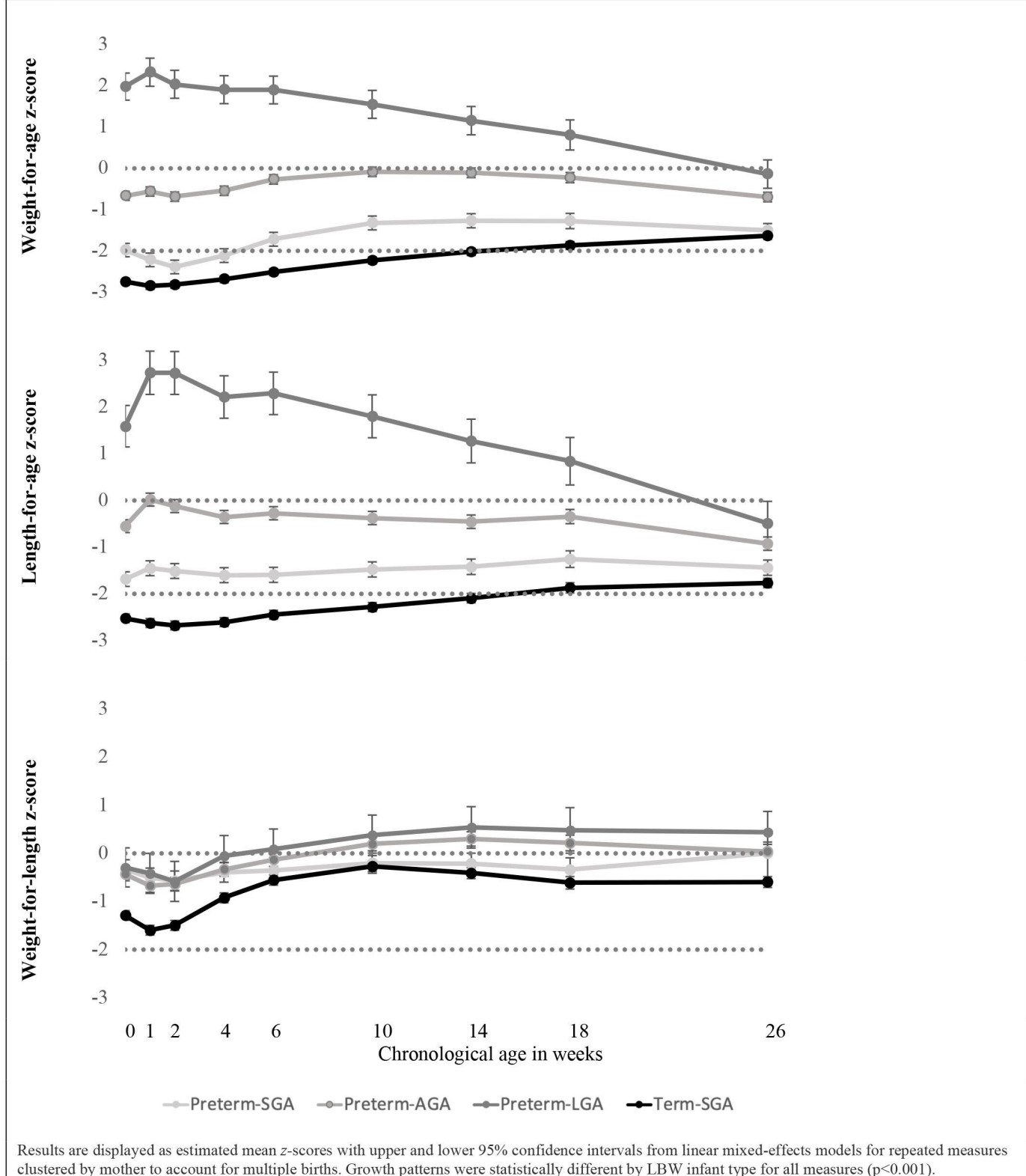

**Figure 2** Unadjusted weight-for-age, length-for-age and weight-for-length z-scores by chronological age for a cohort of preterm-SGA, preterm-AGA, preterm-LGA and term-SGA infants. AGA, appropriate-for-gestational age; SGA, small-for-gestational age; LGA, large-for-gestational age.

infants. Term-SGA infants had 2.33 (95% CI 1.77 to 3.08), 2.89 (95% CI 1.97 to 4.24) and 1.99 (95% CI 1.13 to 3.51) times higher risks of being stunted, underweight and wasted at 6 months, respectively, compared with

preterm-AGA infants. Preterm-LGA was not a risk factor for poor growth outcomes. In terms of growth by chronological age, mean WAZ and LAZ for all LBW types apart from preterm-LGA did not exceed the reference median

**Table 2** Low birth weight type as a predictor for poor growth outcomes at 6 months

| | Prevalence of poor growth outcomes by LBW type | | | | Unadjusted | | | | | Adjusted* | | | | |
|---|---|---|---|---|---|---|---|---|---|---|---|---|---|---|
| | n (%) | | | | | RR (95% CI), p value | | | | | RR (95% CI), p value | | | |
| | Preterm SGA | Preterm AGA | Preterm LGA | Term SGA | N | Preterm SGA | Preterm LGA | Term SGA | N | Preterm SGA | Preterm LGA | Term SGA |
| Stunted at 6 months | 48 (38.1) | 52 (19.9) | 4 (12.9) | 175 (40.1) | 856 | 2.05 (1.49 to 2.83), <0.001 | 0.58 (0.22–1.50), 0.26 | 2.39 (1.82 to 3.15), <0.001 | 831 | 1.89 (1.37 to 2.62), <0.001 | 0.61 (0.23 to 1.60), 0.32 | 2.33 (1.77 to 3.08), <0.001 |
| Underweight at 6 months | 37 (29.4) | 29 (11.1) | 0 (0.0) | 156 (35.5) | 858 | 2.64 (1.70 to 4.08), <0.001 | Non-estimable | 3.18 (2.16 to 4.68), <0.001 | 833 | 2.32 (1.48 to 3.62), <0.001 | Non-estimable | 2.89 (1.97 to 4.24), <0.001 |
| Wasted at 6 months | 12 (9.5) | 14 (5.3) | 0 (0.0) | 62 (14.2) | 857 | 1.56 (0.74 to 3.31), 0.24 | Non-estimable | 2.04 (1.14 to 3.65), 0.02 | 832 | 1.26 (0.57 to 2.79), 0.57 | Non-estimable | 1.99 (1.13 to 3.51), 0.02 |

| | Mean (SD) | | | | Unadjusted | | | | | Adjusted* | | | | |
|---|---|---|---|---|---|---|---|---|---|---|---|---|---|---|
| | Preterm SGA | Preterm AGA | Preterm LGA | Term SGA | | Beta (95% CI) | | | | | Beta (95% CI) | | | |
| | | | | | N | Preterm SGA | Preterm LGA | Term SGA | N | Preterm SGA | Preterm LGA | Term SGA |
| LAZ at 6 months | −1.66 (1.1) | −0.90 (1.3) | −0.47 (1.4) | −1.76 (1.1) | 856 | −0.84 (−1.09 to −0.60), <0.001 | 0.58 (0.06 to 1.10), 0.03 | −1.06 (−1.25 to 0.87), <0.001 | 831 | −0.79 (−1.03 to −0.54), <0.001 | 0.49 (−0.04 to 1.02), 0.07 | −1.05 (−1.24 to 0.86), <0.001 |
| WAZ at 6 months | −1.46 (1.1) | −0.63 (1.1) | −0.11 (0.7) | −1.61 (1.1) | 858 | −0.81 (−1.05 to −0.58), <0.001 | 0.49 (0.19 to 0.78), 0.001 | −0.94 (−1.13 to −0.76), <0.001 | 833 | −0.72 (−0.96 to −0.48), <0.001 | 0.42 (0.12 to 0.73), 0.01 | −0.90 (−1.08 to −0.72), <0.001 |
| WLZ at 6 months | −0.42 (1.2) | 0.06 (1.3) | 0.42 (1.1) | −0.57 (1.3) | 857 | −0.36 (−0.61 to −0.11), 0.01 | 0.17 (−0.26 to 0.59), 0.44 | −0.37 (−0-58 to −0.17), <0.001 | 832 | −0.28 (−0.55 to −0.01), 0.04 | 0.16 (−0.26 to 0.58), 0.44 | −0.32 (−0.53 to −0.10), 0.004 |

Reference group: preterm-AGA infants.

*Adjusted by maternal education, maternal age, parity, place of residence, wealth quintile, birthcount, sex and site; and with cluster-robust SEs for clustering by mother.

AGA, appropriate-for-gestational age; LAZ, weight-for-age; LGA, large-for-gestational age; RR, relative risk; SGA, small-for-gestational age; WAZ, weight-for-age; WLZ, weight-for-length.

**Table 3** Failure to achieve birth weight by 2 weeks as a predictor for poor growth outcomes at 6 months

| | n (%) of poor growth outcomes among infants with no birth weight regain by 2 weeks | Unadjusted | | | Adjusted* | | |
|---|---|---|---|---|---|---|---|
| | | N | RR (95% CI) | P value | N | RR (95% CI) | P value |
| Stunted at 6 months | 72 (40.9) | 838 | 1.39 (1.13 to 1.72) | 0.002 | 819 | 1.51 (1.23 to 1.85) | <0.001 |
| Underweight at 6 months | 62 (34.8) | 839 | 1.46 (1.14 to 1.88) | 0.003 | 821 | 1.55 (1.21 to 1.99) | 0.001 |
| Wasted at 6 months | 20 (11.3) | 838 | 1.04 (0.66 to 1.63) | 0.87 | 820 | 1.08 (0.68 to 1.73) | 0.74 |
| | Mean (SD) z-score among infants with no regain by 2 weeks | N | Beta (95% CI) | P value | N | Beta (95% CI) | P value |
| LAZ at 6 months | −1.70 (1.3) | 837 | −0.38 (−0.61 to −0.16) | 0.001 | 819 | −0.50 (−0.69 to −0.30) | <0.001 |
| WAZ at 6 months | −1.46 (1.2) | 839 | −0.26 (−0.46 to −0.05) | 0.01 | 821 | −0.30 (−0.48 to −0.11) | 0.01 |
| WLZ at 6 months | −0.30 (1.4) | 838 | 0.11 (−0.10 to 0.33) | 0.31 | 820 | 0.12 (−0.09 to 0.34) | 0.26 |

*Adjusted by maternal education, maternal age, parity, place of residence, wealth quintile, birthcount, sex, LBW type, site; and with cluster-robust SEs for clustering by mother.
LAZ, length-for-age; LBW, low birth weight; RR, relative risk; WAZ, weight-for-age; WLZ, weight-for-length.

at any follow-up time point; term-SGA infants had mean WAZ and LAZ scores <−2 from birth through 14 weeks of age (figure 2). The difference in growth patterns by LBW type remained significant in the multivariable model (p<0.001). Similarly, significant differences were found between the four LBW types for mean weight and length by PMA; SGA infants had lower weights and lengths than did those born AGA and LGA (online supplemental figure S1).

Mother–infant dyads who reported feeding difficulties over the first 6 months were more likely to have infants who were underweight at 6 months (RR 1.39, 95% CI 1.09 to 1.78) compared with those never reporting feeding difficulties; no significant associations were observed with 6-month stunted and wasted outcomes (online supplemental table S7). Being stunted, underweight and wasted was more prevalent among infants failing to regain their birth weight by 2 weeks vs those regaining. Mean z-scores for those not regaining birth weight by 2 weeks were <0, lowest for LAZ (table 3). In multivariable models, infants not regaining birth weight by 2 weeks had 1.51 (95% CI 1.23 to 1.85) and 1.55 (95% CI 1.21 to 1.99) times greater risks of being stunted and underweight at 6 months, respectively, compared with infants regaining. There was no significant relationship between lack of birth weight regain by 2 weeks and being wasted at 6 months. Similar trends were observed in multivariable models with continuous outcomes of WAZ, LAZ and WLZ.

Finally, we found no evidence of a relationship between EBF to 6 months and 6 month poor growth outcomes in the multivariable models (stunted 1.08 (95% CI 0.88 to 1.31), underweight 1.08 (95% CI 0.86 to 1.36) and wasted 1.30 (95% CI 0.86 to 1.95)) (online supplemental table S8). Further, this association was not modified by infant birth weight (<2.0 kg vs ≥2.0 to <2.5 kg) (online supplemental table S9).

## DISCUSSION
We found that LBW type and early growth deficits were significant risk factors for poor 6-month growth outcomes. Our LBW infant cohort was heterogeneous with nutritional risks varying by setting and birth outcome. Preterm-SGA and term-SGA infants exhibited greater risks for being stunted and underweight at 6 months compared with preterm-AGA infants. Similarly, infants who failed to regain their birth weight by 2 weeks were more likely to be stunted and underweight at 6 months than those who successfully regained. These findings emphasise the need for early growth monitoring and proactive intervention for small and/or nutritionally at-risk infants. While human milk feeding was the predominant feeding profile among this cohort, 567 (58%) infants were not EBF to 6 months as advised by WHO. Additionally, nearly half of mother–infant dyads reported feeding difficulties in the first 6 months, which reinforces the need for universal and consistent lactation support and management targeted to the needs of LBW infants in low-resource settings. The MAMI Care Pathway Package highlights reported feeding difficulties as criteria for further assessment and enrolment in support at the population level.[12 21]

While a major objective of this study was to fill the gap in evidence regarding moderately LBW infant feeding patterns and their relationship with growth and other health outcomes, we recognise that numerous factors besides feeding impact infant growth including maternal nutritional status, maternal health and well-being, infant illness, intrinsic causes of intrauterine growth restriction,

kangaroo mother care/thermal care and supplements given to infants.[22–29] Prior to this study, little evidence was available on feeding patterns and growth outcomes of moderately LBW infants in LMIC.[30] Differences in risks of poor outcomes by LBW type highlight the need for risk stratification by SGA and LBW type rather than solely by GA or birth weight.[31] Such nuanced risk stratification is rare, but will be featured in the upcoming Lancet Small Vulnerable Newborn Series and other work being done by the WHO.[9] With improved GA dating and increased rates of antenatal care attendance and facility delivery, identification of LBW types will be possible. Our data emphasise term-SGA as an important indicator of risk and highlights the knowledge gap in the timing of their catch-up growth. This finding is supported through the recently published INTERBIO-21st Newborn Study, highlighting increased risk for growth deficits among infants born with intrauterine growth-restriction.[31] Numerous studies have emphasised the poor longer-term growth impact of being born SGA.[32 33]

In order to improve growth patterns and outcomes, early indicators are needed to identify infants in need of close monitoring and timely intervention. In general, growth faltering indicators are not standardised in their definition or use.[34] Lack of birth weight regain by 2 weeks is a common clinical indicator of early feeding and growth problems.[35] However, there is little evidence examining birth weight regain as a predictor of later growth deficits, particularly among LBW infants in LMIC. In the literature, weight gain was more often measured over longer periods and associations with developmental outcomes were explored.[36]

EBF is an optimal source of nutrition for child health and survival, supported by strong evidence of its protective impact on morbidity and mortality.[37 38] However, evidence on the relationship between EBF duration and growth is mixed/conflicting, including similar findings to ours of no known association, modest improvements or deficits.[39 40] This study adds to existing literature by examining this relationship among moderately LBW infants in three LMIC. This complex relationship is not linear and many factors are potentially at play.[41] There is a knowledge gap on how to support optimal growth of vulnerable infants, including assessment of the components of EBF (eg, nutritional composition, volume and feeding frequency), an infant's ability to digest human milk (eg, gut microbiome and enteric infection), maternal nutrition and water, sanitation, and hygiene conditions of the feeding process.[37]

This study had numerous limitations. While the COVID-19 pandemic led to some changes in study activities, study timelines were only slightly impacted and visit attendance remained high. Despite a lack of unified guidance on the application, harmonisation and interpretation of existing child growth standards for this LBW population, we were able to recommend and apply an approach to assess infants against an appropriate standard. We note known difficulties of GA measurement in LMIC and acknowledge the possibility of some misclassification. To identify the best estimate of GA, we used an algorithm based on multiple sources documented in patients' charts. Regardless of possible imprecisions in birth weight measurements, we felt this measure taken within minutes/hours of birth was more accurate than using enrolment weight (taken in triplicate by study nurses using standardised, calibrated scales) as a proxy of birth weight; a sensitivity analysis was conducted using enrolment weight in the models and similar trends were observed. Possible response bias was minimised through data collection by research staff not known to respondents rather than their care providers. As in every longitudinal study, missing data and lost to follow-up was present; we tried to minimise this by conducting phone interviews or extending visit windows where in-person visits were not possible. Exclusion of LBW infants with congenital abnormalities that interfere with feeding at the screening stage of the study means that the particular needs and likely poorer health outcomes associated with early life disability are not captured in this paper. Finally, when determining generalisability of the findings, it should be noted that our cohort had ready access to secondary/tertiary health facilities; and the majority of preterm infants were late-preterm, thus influencing feeding patterns/abilities.

## CONCLUSION

This study examined the feeding and growth patterns of moderately LBW infants in the first half of infancy. Our LBW infant cohort was heterogeneous, comprising four LBW types that differed in prevalence by region. LBW type, particularly those born SGA regardless of preterm or term status, and lack of birthweight regain by 2 weeks may be important parameters that could be used to identify and proactively manage nutritionally at-risk infants early in life. Additionally, utilisation of these indicators could facilitate the prioritisation of scarce resources (eg, facility staff, space and breast pumps) and services (eg, lactation support) to infants at highest risk. Research is needed to support optimal feeding strategies for LBW infants, understand the needs and growth patterns of SGA infants and evaluate the role of human milk volume, nutrient composition and feeding frequency on infant growth.

**Author affiliations**
[1]Ariadne Labs, Harvard T.H. Chan School of Public Health / Brigham and Women's Hospital, Boston, Massachusetts, USA
[2]Jawaharlal Nehru Medical College, KLE Academy of Higher Education and Research (Deemed-to-be-University), Belgaum, Karnataka, India
[3]Department of Pediatrics and Child Health, Muhimbili University of Health and Allied Sciences, Dar es Salaam, Tanzania
[4]University of North Carolina Project Malawi, Lilongwe, Malawi
[5]Department of Epidemiology and Population Health, Stanford University, Palo Alto, California, USA
[6]Department of Global Health and Population, Harvard T.H. Chan School of Public Health, Boston, Massachusetts, USA
[7]Institute for Global Health and Infectious Diseases, University of North Carolina at Chapel Hill School of Medicine, Chapel Hill, North Carolina, USA

[8]Department of Pediatrics, University of North Carolina at Chapel Hill School of Medicine, Chapel Hill, North Carolina, USA
[9]Department of Paediatrics, SCB Medical College & Hospital, Cuttack, Orissa, India
[10]Department of Paediatrics, JJM Medical College, Davangere, Karnataka, India
[11]Department of Paediatrics, City Hospital, Cuttack, Orissa, India
[12]Department of Paediatrics, SS Institute of Medical Sciences and Research Center, Davangere, Karnataka, India
[13]Department of Nutrition, University of North Carolina at Chapel Hill, Gillings School of Global Public Health, Chapel Hill, North Carolina, USA
[14]Hubert Department of Global Health, Emory University School of Public Health, Atlanta, Georgia, USA
[15]Center for Nutrition, Boston Children's Hospital, Boston, Massachusetts, USA
[16]Maternal, Newborn, Child Health and Nutrition Program, PATH, Seattle, Washington, USA
[17]Department of Pediatric Newborn Medicine, Brigham and Women's Hospital, Boston, Massachusetts, USA
[18]Harvard Medical School, Boston, Massachusetts, USA

**Acknowledgements** The authors would like to thank clinical leadership and staff at all study facilities for their partnership, support and contribution to this work; the mothers and infants for allowing us to have a glimpse into their experiences and sharing key moments of their lives; and all data collectors and study staff for conducting study activities.

**Contributors** Study conceptualisation and design was completed by ACCL, BAC, CD, CRS, DET, IH, KEAS, KB, KLM, KMa, KMi, KN, LA, LS, LV, MY, RMB, SD, SSG, SLM and TM. Protocol development and tool design was carried out by ACCL, BAC, CRS, CD, DET, EF, IH, KEAS, KF, K-IB, KLM, KMa, KMi, KN, LA, LD, LS, LV, MY, NS, RMB, RK, SD, SSG, SLM, SM, SP, SS, SSV and TM. Tool pilot testing/ modification were conducted by DET, EF, FS, KF, MBK, LD, LGS, LS, LV, MP, MS, NS, RK, SM, SS, SSV, VH, YM. Data collection, cleaning and management was carried out by AP, EV, FN, MBK, KMs, LD, LS, LV, MB, KL, SP, YK. Analysis and interpretation were conducted by CRS, DET, EF, EV, GB, KEAS, KMi, LA, LV, RRM and SL. Writing of the manuscript was done by LV. All authors reviewed the study results and multiple drafts of the manuscript and approved the final version. LV acted as the guarantor of the manuscript.

**Funding** This publication is based on research funded in part by the Bill & Melinda Gates Foundation. The findings and conclusions contained within are those of the authors and do not necessarily reflect positions or policies of the Bill & Melinda Gates Foundation (INV-007326).

**Competing interests** All authors completed the ICMJE conflict of interest form and were funded by the Bill & Melinda Gates Foundation for this work as part of the LIFE study. ACCL, BAC, CD, CRS, DET, KEAS, K-IB, KLM, KM, MY have received funding from the Bill & Melinda Gates Foundation for maternal and newborn health and nutrition work at large. CD reports other funding from American Society for Nutrition, UpToDate and People's Medical Publishing House. ACCL reports grants from the WHO and National Institute of Health/ NICHD. BAC reports funding from UNICEF and the US National Health Institutes of Health. MY reports grants from NIH, Emory University, and the Centers for Disease Control. K-IB and KM report grants from the Philips Foundation, the WHO and USAID. SLM has received funding from the International Society for Research on Human Milk and Lactation. All other authors have declared no conflicts of interest.

**Patient and public involvement** Patients and/or the public were involved in the design, or conduct, or reporting, or dissemination plans of this research. Refer to the Methods section for further details.

**Patient consent for publication** Consent was obtained directly from patient(s).

**Ethics approval** This study was approved by 11 ethics committees in India, Malawi, Tanzania and the USA: (1) India Health Ministry's Screening Committee with Indian Council of Medical Research acting as its secretariat (2019-2674); (2) Directorate of Health and Family Welfare Services, Government of Karnataka which also covers investigators at Women and Children Hospital, Davangere and Chigateri General District Hospital, Davangere (NHM/SPM/04/2019-20); (3) Institutional Ethics Committee of KLE Academy of Higher Education and Research which also covers investigators at JN Medical College, Belagavi and KLES Dr Prabhakar Kore Hospital & Medical Research Center, Belagavi (KAHER/IEC/2019-20/D-2760); (4) Institutional Ethics Review Board of SS Institute of Medical Sciences & Research Centre (IERB/200/2019); (5) Institutional Ethics Committee of JJM Medical College (JJMMC/IEC-01/2019) which also covers investigators at Bapuji Child Health Institute & Research Centre, Davangere, Women & Children Hospital, Davangere and Chigateri General District Hospital, Davangere; (6) Research and Ethics Committee, Directorate of Health Services, Odisha State which also covers investigators at City Hospital Oriya Bazar, Cuttack (155/PMU/187/17); (7) Institutional Ethical Committee, Sriram Chandra Bhanja Medical College, Cuttack (7188); (8) Malawi National Health Sciences Research Committee (NHSRC2019/Protocol19/03/2250-UNCPM 21905); (9) Tanzania National Institute of Medical Research (NIMR/HQ/R.8a/Vol.IX/3126); (10) Muhimbili University of Health and Allied Sciences (DA.282/298/01.C/); and (11) USA - Harvard T.H Chan School of Public Health (IRB10-0282) which also covers investigators at Boston Children's Hospital, Brigham and Women's Hospital, Emory University, PATH and University of North Carolina.

**Provenance and peer review** Not commissioned; externally peer reviewed.

**Data availability statement** Data are available in a public, open access repository. Deidentified individual participant data (including data dictionaries) used in this manuscript are available through the Harvard Dataverse platform under the BetterBirth Dataverse website. This can be found at: https://dataverse.harvard.edu/dataverse/BetterBirthData.

**Author note** The reflexivity statement for this paper is linked as an online supplemental file 3.

**ORCID iDs**
Linda Vesel http://orcid.org/0000-0003-3753-4172
Karim Manji http://orcid.org/0000-0002-7069-6408
Friday Saidi http://orcid.org/0000-0003-1190-1499
Christopher R Sudfeld http://orcid.org/0000-0002-3203-3638
Sangappa Dhaded http://orcid.org/0000-0001-9575-5251
Anne C C Lee http://orcid.org/0000-0003-2654-9862
Katherine E A Semrau http://orcid.org/0000-0002-8360-1391

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
