## [Reviewer comments · BMJ Open]

ARTICLE DETAILS

TITLE (PROVISIONAL)	Feeding Practices and Growth Patterns of Moderately Low Birthweight Infants in Resource-limited Settings: Results from a Multi-site, Longitudinal Observational Study
AUTHORS	Vesel, Linda; Bellad, Roopa; Manji, Karim; Saidi, Friday; Velasquez, Esther; Sudfeld, Christopher; Miller, Katharine; Bakari, Mohamed; Lugangira, Kristina; Kisenge, Rodrick; Salim, Nahya; Somji, Sarah; Hoffman, Irving; Msimuko, Kingsly; Mvalo, Tisungane; Nyirenda, Fadire; Phiri, Melda; Das, Leena; Dhaded, Sangappa; Goudar, Shivaprasad S.; Herekar, Veena; Kumar, Yogesh; Koujalagi, M B; Guruprasad, Gowdar; Panda, Sanghamitra; Shamanur, Latha; Somannavar, Manjunath; Vernekar, Sunil; Misra, Sujata; Adair, Linda; Bell, Griffith; Caruso, Bethany; Duggan, Christopher; Fleming, Katelyn; Israel-Ballard, Kiersten; Fishman, Eliza; Lee, Anne; Lipsitz, Stuart; Mansen, Kimberly; Martin, Stephanie; Mokhtar, Rana; North, Krysten; Pote, Arthur; Spigel, Lauren; Tuller, Danielle; Young, Melissa; Semrau, Katherine

VERSION 1 – REVIEW

REVIEWER	Abdallah, Yaser Makerere University, Paediatrics and Child Health
REVIEW RETURNED	30-Aug-2022

GENERAL COMMENTS	Review comments Thank you for taking time to describe feeding and growth patterns and early risk factors for growth faltering. The objectives are clear but the variables to be considered including maternal nutritional status, sick infant status in the 6months follow up period, intrinsic causes of intrauterine growth restriction, kangaroo/thermal care in the first few weeks, supplement's given to infants including multivitamin and iron. These variables need to be factored in besides feed to explain growth!! The Introduction is precise though paragraph one intending to state that LBW infants may be preterm or term, SGA or AGA is not coming up clearly. Paragraph 2 reference 6 this review has studies predominantly from high income countries, this should come out clearly. Paragraph 3 for justification should not use COVID 19 any anyway since study started enrolment pre COVID pandemic. Of note is 67.3% of study infants were SGA, a detailed description of this population is lacking, where they symmetrically growth restricted? Although anthropometry where done, there is no mention on head circumferences!! My observation is 67.3% were SGA at birth only 25.8%(222/858) were underweight at 6months!! Does this
---

	signal catchup growth in a good proportion??? Since the study infants were assessed on day 0-72hrs, day 7, day 14, what interventions were instituted among those faltering on growth early? Where they just observed and declared failed to recover birth weight?? The randomization process, it is apparent (table 1) that study infants from India were predominantly term SGA 70% compared to Africa with majority being preterms. This may affect generalization of the findings since the care in these 2 continents are different There is n clear description of growth of those exclusively breast fed and those exclusively formular fed, a comparison would be relevant. It seems feeding difficulty was common (page 11, line 32) 44.3% although this should be described in detailed especially with respect to time was 44.3% in the first review, 2nd, 3rd..., this is relevant considering a good majority got some form of lactation review/guidance. Line 34 difficulty descriptions long time for milk to come and “milk let down” it is not clear if milk let down is a problem or they meant no milk let down?? Association between feeding difficulty and growth faltering is worthy analysis Supplementary material page 21 the curves have no reference curve to compare?? There is need to compute growth velocity They must show head growth since this may have neurodevelopmental implications
--	---

REVIEWER	McGrath, Marie Emergency Nutrition Network
REVIEW RETURNED	30-Sep-2022

GENERAL COMMENTS	Abstract Lines 29-31: Abstract conclusion: Suggest to nuance more to reflect that these are not the only care components needed (eg clinical care and maternal nutrition/health interventions may also be warranted) that growth monitoring in itself won't make a difference without being acted upon, and that bot early weight gain (to regain birth weight) and then sustained growth are needed: Early interventions are needed that include optimal feeding support and action-oriented growth monitoring to enable appropriate weight gain and proactive management of vulnerable infants. Introduction Lines 23 to 28. Also important to note is a global implementation guidance (MAMI Care Pathway), collectively developed by the MAMI Global Network, targets SMALL and/or nutritionally at risk infants under 6 months of age and their mothers, including LBW as a criteria for enrollment into integrated nutrition and health care. This collective effort has, for the past 10 years or more, sought to highlight the need for and to build actionable evidence to improve
--

	quality of care provision and to prevent and manage poor outcomes in infants u6m including LBW infants and their mothers. https://www.enonline.net/mamicarepathway Lines 39-41: The authors state that knowledge will inform feeding interventions. However, poor growth may be also due to clinical or maternal factors and feeding interventions are key but will not likely be sufficient for all. Suggest to nuance this so as to avoid reader interpreting this as feeding interventions to address poor growth are the only intervention needed. Methods Lines 50-52. In terms of infants excluded, is this data available for analysis/was there any follow up of these infants? There is a lack of outcome data on infants with disability and as a result a lack of visibility and guidance on their management. This was raised in recent WHO guideline development group meeting in Geneva and reflected in publication recently. Not related to this paper but as a bonus activity, I encourage the authors to analyse data of available or share with WHO (who are undertaking a pooled data analysis to stratify risk in infants and children). Engl, M., et al. (2022). "Children living with disabilities are neglected in severe malnutrition protocols: a guideline review." Archives of Disease in Childhood 107(7): 637-643. Lines 14-15: Stunted/stunting and wasting/wasted terms used interchangeably. Suggest to use wasting or stunting when referring to concept or process and wasted/stunted when referring to measures at timepoints. For example: Current: Six-month stunting, underweight and wasting were defined by z-scores <-2SD for length-for-age (LAZ), weight-for-age (WAZ) and weight-for-length (WLZ), respectively. Possible revision: Six-month stunted, underweight and wasted were defined by z-scores <-2SD for length-for-age (LAZ), weight-for-age (WAZ) and weight-for-length (WLZ), respectively. Lines 15-16: Defines SAM as WLZ <-3. Is there a reason for using SAM terminology here rather than 'severe wasting' since this is the first time it referred to, and it is 'wasting' rather than 'SAM' that you explore in the analysis. Results Lines 14-18: Was bilateral oedema checked? Line 32 What was correlation between reported feeding difficulties and growth outcomes? Lines 3-5: How did Term SGA v Preterm SGA compare in terms of risk of poor growth outcomes? Lines 40-41: Suggest as per comment on terminology above; Current: Stunting, underweight, and wasting were more prevalent among infants failing to regain their birthweight by two weeks versus those regaining. Revised: Stunted, underweight, and wasted were more prevalent
--	--

among infants failing to regain their birthweight by two weeks versus those regaining.

Lines 42 and 44: Suggest as per comment on terminology above;

Current: In multivariable models, infants not regaining birthweight by two weeks had 1.51 (1.23-1.85) and 1.55 (1.21-1.99) times greater risks of six-month stunting and underweight, respectively, compared to infants regaining.

Revised: In multivariable models, infants not regaining birthweight by two weeks had 1.51 (1.23-1.85) and 1.55 (1.21-1.99) times greater risks of six-month stunted and underweight, respectively, compared to infants regaining.

Lines 15-19: It would be valuable to know the association between reported feeding difficulties and six month poor growth outcomes.

Lines 25-27:

Current: Preterm-SGA and term-SGA infants exhibited greater risks for six-month stunting and underweight compared to preterm-AGA infants.

Revised: Preterm-SGA and term-SGA infants exhibited greater risks for six-month stunted and underweight compared to preterm-AGA infants.

Discussion

Lines 28-29: Suggest refine to include 'early' and to refer to 'small' as well as nutritionally at risk as some LBW infants may not initially be identified as wasted/stunted/underweight/growth faltering.

Suggested revision to read:

These findings emphasize the need for early growth monitoring and proactive intervention for small and/or nutritionally at-risk infants.

Lines 29-30. It would also be valuable to discuss the association between reported feeding difficulties and poor growth outcomes at 6 months. Nearly half of all mothers reported feeding difficulties. The MAMI Care Pathway, referred to earlier, includes reported feeding difficulties as criteria for further assessment and enrollment in support.

Lines 35-37: Note that risk stratification by gestational age is a feature of pooled data analysis underway by WHO that would be useful to refer to here. More details are available from Nigel Rollins, WHO.

Lines 37-39: Improved GA dating, etc will only be available in limited LMICs. In reality, capacity to identify LBW, never mind sub--types, will be limited/non-existent. I think an additional reflection (and a key learning from this paper) could be that it is very important for those running prevention and/or treatment services for wasting, stunting and underweight recognise that they are dealing with at risk infants and children with differing birth histories and early growth trajectories that have (and will continue) to contribute to poor growth (and most importantly, poor functional) outcomes in both

early and later life. There needs to be much more integrated guidance and continuity of care provision between reproductive/maternal/neonatal health services that identify higher risk infants and their mothers, and subsequent community based interventions and active responsive growth surveillance.

Lines 4-8: These are important factors but centre on specific quite technical components of feeding and does not take into account the wider determinants/interactions on growth and requiring a more holistic/cross service interventions. Nearly half of mothers reported feeding difficulties. It would be important to research further what type of feeding support they got (the study reports in general terms quite a range from individual to group sessions). Other important (and highly relevant) factors will include clinical care, maternal nutrition and health including mental wellbeing, and the role of wider family and community support to mothers in their central role in young infant growth and development.

Lines 25-26: An additional valuable reflection may be that exclusion of infants with congenital abnormalities, while justified for study purposes, does mean that the particular needs and likely poorer growth outcomes associated with early life disability is not captured in this analysis.

Lines 44 and 45: Early indicators are need but also close collaboration between early postnatal care and subsequent surveillance and follow up in the community for LBW infants.

Lines 44-46: Suggest to use phrase 'growth faltering' rather than growth failure, which is more consistent with the approach reflected in the paper and analysis.

Hoehn C, Lelijveld N, Mwangome M, Berkley JA, McGrath M, Kerac M. Anthropometric Criteria for Identifying Infants Under 6 Months of Age at Risk of Morbidity and Mortality: A Systematic Review. *Clin Med Insights Pediatr.* 2021 Oct 21;15:11795565211049904. doi: 10.1177/11795565211049904. PMID: 34707425; PMCID: PMC8543668.

Note also this paper:

Mwangome, M., et al. (2019). "Anthropometry at birth and at age of routine vaccination to predict mortality in the first year of life: A birth cohort study in BukinaFaso." *PLoS One* 14(3): e0213523.

Mwangome, M., et al. (2021). "Growth monitoring and mortality risk in low birthweight infants: a birth cohort study in Burkina Faso [version 1; peer review: 1 approved with reservations]." *Gates Open Research* 5(82).

Conclusions

Lines 33-34: Regarding the following statement:

LBW type and early growth may be important parameters that could be used to identify and proactively manage nutritionally at-risk infants.

There is a risk that the emphasis on identification of LBW type is necessary to identify and manage at risk infants. Whilst ideal, this

	pursuit of the ideal may be the enemy of the good. This study shows that many of these particularly at risk LBW types are 'captured' by anthropometric indicators, such as underweight (which is a feasible, well established and evidenced indicator of risk of poor outcomes). For example, note systematic review that indicates weight for age, followed by MUAC, as indicators that identify infants u6m at risk of poor outcomes. For example: Hoehn C, Lelijveld N, Mwangome M, Berkley JA, McGrath M, Kerac M. Anthropometric Criteria for Identifying Infants Under 6 Months of Age at Risk of Morbidity and Mortality: A Systematic Review. Clin Med Insights Pediatr. 2021 Oct 21;15:11795565211049904. doi: 10.1177/11795565211049904. PMID: 34707425; PMCID: PMC8543668. Note also this paper: Also: Mwangome, M., et al. (2019). "Anthropometry at birth and at age of routine vaccination to predict mortality in the first year of life: A birth cohort study in BukinaFaso." PLoS One 14(3): e0213523. Given that, I think the conclusions should reflect this, perhaps noting that early identification of growth problems in infants under 6 months will identify higher risk infants that warrant close surveillance and prompt support. Infants who are known to be small at birth should be automatically enrolled in supportive care. Such an approach is reflected in the MAMI Care Pathway approach, referred to earlier. Lines 34-38: Regarding the following conclusions: Additionally, utilization of these indicators could facilitate the prioritization of scarce resources (e.g., facility staff, space, and breast pumps) and services (e.g., lactation support) to infants at highest risk. Research is needed to support optimal feeding strategies for LBW infants and evaluate the role of human milk volume, nutrient composition, and feeding frequency on infant growth. In practice, early growth patterns and anthropometric indicators and birth history are more likely to be feasible compared to identifying sub-types of LBW (not detracting from the value these would have where there is resource and capacity to measure these). Throughout the paper there is no mention of the role of the mother's nutrition, health and wellbeing in determining growth outcomes of her infant. Some data on the mother is gathered, eg age, but there is no reflection on this. Appreciating the focus is on infant parameters and therefore conclusions are accordingly, this could perhaps be included in the limitations section. Figure 1: Amongst infants excluded, quite a number were excluded as 6-month follow-up not feasible 745 (72%). What were the factors that made it not feasible? Tables: Note suggested revision to terminology (e.g. stunted rather than stunting at 6 months).
--	--

Reviewer #1 - Miss Yaser Abdallah, Makerere University

Comment #1: Thank you for taking time to describe feeding and growth patterns and early risk factors for growth faltering.

Response #1: Thank you for your review. We appreciate your time.

Comment #2: The objectives are clear but the variables to be considered including maternal nutritional status, sick infant status in the 6months follow up period, intrinsic causes of intrauterine growth restriction, kangaroo/thermal care in the first few weeks, supplement's given to infants including multivitamin and iron. These variables need to be factored in besides feed to explain growth!!

Response #2: Thank you for making this point and we absolutely agree. We have added the following text to the beginning of the second paragraph in the discussion and have added references to support it: "While a major objective of this study was to fill the gap in evidence regarding moderately LBW infant feeding patterns and their relationship with growth and other health outcomes, we recognize that numerous factors besides feeding impact infant growth including maternal nutritional status, infant illness, intrinsic causes of intrauterine growth restriction, kangaroo mother care/thermal care and supplements given to infants."

Comment #3: The Introduction is precise though paragraph one intending to state that LBW infants may be preterm or term, SGA or AGA is not coming up clearly. Paragraph 2 reference 6 this review has studies predominantly from high income countries, this should come out clearly. Paragraph 3 for justification should not use COVID 19 anyway since study started enrolment pre COVID pandemic.

Response #3: Thank you for the detailed feedback. In paragraph 1, we have made the sentence that defines LBW infants into two sentences to create clarity. In paragraph 2, we have clarified that the existing evidence is concentrated in high-income countries. Finally, in paragraph 3, we added the statement about COVID to put the importance of this work into context and point out its relevance given the current public/ clinical health environment. We have added that the focus on vulnerable infant care is "even more" critical, which we hope will clarify our intention with this statement. Additionally, COVID impacted enrollment as we had to pause enrollment in Malawi and India Odisha at the height of the pandemic; further, we followed these infants and their families throughout the pandemic.

Comment #4: Of note is 67.3% of study infants were SGA, a detailed description of this population is lacking, were they symmetrically growth restricted? Although anthropometry where done, there is no mention on head circumferences!! My observation is 67.3% were SGA at birth only 25.8% (222/858) were underweight at 6months!! Does this signal catchup growth in a good proportion???

Response #4: We show the growth patterns of each LBW type in Figure 2 and supplementary Figure 1. We have explained both figures in the results section. Although they do grow slowly over time, it is clear that SGA infants are worse off than AGA infants in terms of where they start off and where they end up as well as exhibiting suboptimal growth over time. We did measure head circumference, but concentrated this paper on weight and length as they are the most commonly accessible metrics. We will share findings related to head circumference and mid-upper arm circumference in forthcoming publications. Overall, 25.8% of infants were underweight at 6 months; this is a large burden that exceeds national, regional and LMIC estimates (<https://data.worldbank.org/indicator/SH.STA.MALN.ZS>). As shown in Table 2, the prevalence of underweight and other poor growth outcomes was higher among preterm SGA and term SGA infants compared to preterm AGA infants.

Comment #5: Since the study infants were assessed on day 0-72hrs, day 7, day 14, what interventions were instituted among those faltering on growth early? Where they just observed and declared failed to recover birth weight??

Response #5: While this is an observational study without an intervention arm, we maintained an ethical obligation to study participants and their families to connect them with local resources available to manage any harmful/ concerning situations that may have arisen. We employed a safety net standard operating procedure that included the completion of a safety net module by data collectors/ study staff at all study visits. Indicators we assessed included: malnutrition / severe growth

faltering (failure to regain birthweight by 4 weeks, weight-for-length <-3 z-scores, oedema in both feet, visible wasting), danger signs or severe illness among the mother or infant. When we encountered an infant/family with any indication, a subsequent referral for advanced care was completed. We have added the above text into the last paragraph of the procedures section of the methods.

Comment #6: The randomization process, it is apparent (table 1) that study infants from India were predominantly term SGA 70% compared to Africa with majority being preterms. This may affect generalization of the findings since the care in these 2 continents are different.

Response #6: We agree with your assessment, and it is true that there are major regional differences in LBW type as well as other outcomes. This has been noted by others including Lee 2013 using 2010 data from 138 LMICs, namely SGAs were predominant in Asia and preterm infants in Africa. As this was an observational study, we did not conduct any randomization procedures; here, we enrolled all LBW infants born or referred (within the first 72 hours of life) to the 12 study facilities over a set time period. We do not intend to generalize region-specific findings and implications to other regions but rather to make general statements regarding the pooled data. One of our main findings is that moderately LBW infants are a heterogenous group.

Comment #7: There is no clear description of growth of those exclusively breastfed and those exclusively formula fed, a comparison would be relevant.

Response #7: Thank you for the comment; we have updated our analysis to address the concern. We did not have any infants in our cohort who were exclusively formula fed. Most non-exclusively breastfed infants received a mix of breastmilk and something else (i.e., formula, water, other liquids, or semi-solid/solid foods later on). We have added an extra column (#3) to Table S8 in the Supplementary Tables PDF (previously labeled as S7) with data on “n (%) of poor growth outcomes among infants not-exclusively breastfed to 6 months.” Please also see the revised table below for easy reference.

Table S8. Duration of exclusive breastfeeding and growth outcomes

	Exclusive breastfeeding to 6 months or duration of exclusive breastfeeding (in weeks)							
	n (%) of poor growth outcomes among infants		Unadjusted			Adjusted*		
	Exclusively breastfed to 6 months	Non-exclusively breastfed to 6 months	N	RR/Beta (95% CI)	p-value	N	RR/Beta (95% CI)	p-value
Stunted at 6 months	116 (32.0%)	158 (32.9%)	847	1.00 (0.81-1.24)	0.97	821	1.08 (0.88-1.31)	0.47
Underweight at 6 months	91 (25.1%)	127 (26.3%)	849	0.98 (0.77-1.26)	0.90	823	1.08 (0.86-1.36)	0.51
Wasted at 6 months	40 (11.0%)	45 (9.3%)	848	1.15 (0.77-1.73)	0.50	822	1.30 (0.86-1.95)	0.22
	Mean (SD) z-score among infants		Unadjusted			Adjusted*		
	Exclusively breastfed to 6 months	Not exclusively breastfed to 6 months	N	Beta (95% CI)	p-value	N	Beta (95% CI)	p-value
LAZ at 6 months	-1.38 (1.32)	-1.45 (1.23)	847	0.04 (-0.14-0.22)	0.67	821	-0.03 (-0.19-0.14)	0.74
WAZ at 6 months	-1.20 (1.24)	-1.26 (1..18)	849	0.03 (-0.15-0.20)	0.76	823	-0.07 (-0.23-0.09)	0.37
WLZ at 6 months	-0.33 (1.31)	-0.32 (1.32)	848	-0.04 (-0.22-0.14)	0.69	822	-0.09 (-0.27-0.09)	0.31

*Adjusted for maternal education, maternal age, parity, wealth quintile, residence, infant sex, birthcount, and LBW type; and with cluster-robust standard errors for clustering by mother.

Comment #8: It seems feeding difficulty was common (page 11, line 32) 44.3% although this should be described in detail especially with respect to time was 44.3% in the first review, 2nd, 3rd..., this is relevant considering a good majority got some form of lactation review/guidance. Line 34 difficulty

descriptions long time for milk to come and “milk let down” it is not clear if milk let down is a problem or they meant no milk let down??

Response #8: While 77.1% of mother-infant dyads received lactation support or management once during the first 3 days of life, it was not given to all infants and was not comprehensive or consistent throughout the hospital stay (based on findings from a smaller in-facility observational cohort to be published shortly in *PLOS Global Health*). All mothers, particularly those with LBW infants, should receive lactation support; coverage should be 100% in the first few days of life. Women identified the feeding problems they were experiencing, including slow milk production. By “milk let down,” we mean “no milk let down” as you mentioned and have made the change in the text. In addition, we added data on feeding difficulties reported over several timeframes (0-2 weeks, 0-4 weeks and 6-26 weeks) into Table S1 in the Supplemental Tables PDF. Please see the revised Table S1 below.

Table S1. Feeding initiation, exclusivity, difficulties, and counseling

n (%) infants for whom breastfeeding was initiated within 1 hour of birth	Preterm SGA N=124	Preterm AGA N=265	Preterm LGA N=33	Term SGA N=551	Overall N=973
	29 (23.4%)	64 (24.2%)	7 (21.2%)	263 (47.7%)	363 (37.3%)
n (%) infants exclusively breastfed to 4 months	Preterm SGA N=137	Preterm AGA N=282	Preterm LGA N=32	Term SGA N=517	Overall N=968
	79 (57.7%)	177 (62.8%)	25 (78.1%)	295 (57.1%)	576 (59.5%)
n (%) infants exclusively breastfed to 6 months	Preterm SGA N=132	Preterm AGA N=277	Preterm LGA N=31	Term SGA N=531	Overall N=971
	62 (47.0%)	128 (46.2%)	13 (41.9%)	222 (41.8%)	425 (43.8%)
n (%) infants who mothers reported feeding difficulties* during first 6 months	Preterm SGA N=151	Preterm AGA N=327	Preterm LGA N=37	Term SGA N=597	Overall N=1112
	81 (53.6%)	153 (46.8%)	16 (43.2%)	243 (40.7%)	493 (44.3%)
n (%) infants who mothers reported feeding difficulties* during first 2 weeks	Preterm SGA N=151	Preterm AGA N=327	Preterm LGA N=37	Term SGA N=597	Overall N=1112
	55 (36.4%)	100 (30.6%)	11 (29.7%)	137 (22.9%)	303 (27.2%)
n (%) infants who mothers reported feeding difficulties* during first 4 weeks	Preterm SGA N=129	Preterm AGA N=283	Preterm LGA N=32	Term SGA N=509	Overall N=953
	59 (45.7%)	109 (38.5%)	11 (34.4%)	155 (30.5%)	304 (31.9%)
n (%) infants who mothers reported feeding difficulties* from 6 week to 6 months	Preterm SGA N=151	Preterm AGA N=327	Preterm LGA N=37	Term SGA N=597	Overall N=1112
	48 (31.8%)	77 (23.5%)	6 (16.2%)	163 (27.3%)	294 (26.4%)
n (%) infants who received feeding counseling/support at baseline†	Preterm SGA N=151	Preterm AGA N=327	Preterm LGA N=37	Term SGA N=594	Overall N=1109
	118 (78.2%)	267 (81.7%)	27 (73.0%)	443 (74.6%)	855 (77.1%)

*Feeding difficulties included perception of insufficient milk, distracted baby, long time for milk to come in, no milk let down, breast pain, trouble sucking/latching on, etc.
 †Within 72 hours of birth.

Comment #9: Association between feeding difficulty and growth faltering is worthy analysis

Response #9: Thank you for this suggestion. We conducted the suggested analyses and have added the results to a new supplementary table (S7) and the text in the results section after Figure 2. Please see Table S7 below for easy reference.

Table S7. Feeding difficulties in the first six months and poor growth outcomes at six months

	n (%) of poor growth outcomes among infants whose mothers ever reported feeding difficulties in first 6 months of the infant's life	Unadjusted			Adjusted*		
		N	RR (95% CI)	p-value	N	RR (95% CI)	p-value
Stunted at 6 months	153 (37.1%)	858	1.21 (0.99-1.48)	0.063	833	1.13 (0.92-1.39)	0.227
Underweight at 6 months	127 (30.6%)	860	1.55 (1.22-1.98)	<0.001	835	1.39 (1.09-1.78)	0.01
Wasted at 6 months	46 (11.1%)	859	1.49 (1.00-2.21)	0.05	834	1.36 (0.91-2.03)	0.134

*Adjusted by maternal education, maternal age, parity, place of residence, wealth quintile, birthcount, sex, LBW type, site; and with cluster-robust standard errors for clustering by mother.

Comment #10: Supplementary material page 21, the curves have no reference curve to compare??

Response #10: That is correct. Supplemental figure 1 is showing mean weight and mean length patterns over time by postmenstrual age. Unlike Figure 2, it is merely meant to depict and report differences between these patterns by LBW type. In this figure, growth patterns were statistically different by LBW type for both measures. Conversely, the curves in Figure 2 were created using the INTERGROWTH-21st and WHO standards which describe optimal growth for healthy preterm or term infants respectively. In the figure, we used dotted lines to denote 0 and -2 z-scores; below zero signifies suboptimal and <-2 is defined as stunted, wasted or underweight. There is unfortunately no normal birthweight comparison group in our study.

Comment #11: There is need to compute growth velocity. They must show head growth since this may have neurodevelopmental implications

Response #11: We did compute growth velocity and collect head circumference as part of the study and will use this information in a separate publication. In this paper, we focus on reporting findings related to weight and length, common indicators of attained size. We have added a note about this in the second paragraph of the procedures section in the methods.

Reviewer #2 - Mrs. Marie McGrath, Emergency Nutrition Network

Comment #1: Abstract - Lines 29-31: Abstract conclusion: Suggest to nuance more to reflect that these are not the only care components needed (eg clinical care and maternal nutrition/health interventions may also be warranted) that growth monitoring in itself won't make a difference without being acted upon, and that bot early weight gain (to regain birth weight) and then sustained growth are needed.

Early interventions are needed that include optimal feeding support and action-oriented growth monitoring to enable appropriate weight gain and proactive management of vulnerable infants.

Response #1: Thank you for the comment and recommended revision. You are absolutely correct that this sentence should be more nuanced. We have made the change as suggested.

Comment #2: Introduction - Lines 23 to 28. Also important to note is a global implementation guidance (MAMI Care Pathway), collectively developed by the MAMI Global Network, targets SMALL

and/or nutritionally at risk infants under 6 months of age and their mothers, including LBW as a criteria for enrollment into integrated nutrition and health care. This collective effort has, for the past 10 years or more, sought to highlight the need for and to build actionable evidence to improve quality of care provision and to prevent and manage poor outcomes in infants u6m including LBW infants and their mothers. <https://www.enonline.net/mamicarepathway>

Response #2: Thank you for highlighting this. We have added the following sentence to the last paragraph of the introduction: "The MAMI Care Pathway group has focused on highlighting the need for actionable evidence to improve quality of care provision and to prevent and manage poor outcomes in infants in the first six months of life, including LBW infants and their mothers (<https://www.enonline.net/attachments/4004/MAMI-Care-Pathway-Package-Document-07June2021.pdf>)."

Comment #3: Introduction - Lines 39-41: The authors state that knowledge will inform feeding interventions. However, poor growth may be also due to clinical or maternal factors and feeding interventions are key but will not likely be sufficient for all. Suggest to nuance this so as to avoid reader interpreting this as feeding interventions to address poor growth are the only intervention needed.

Response #3: We have edited the sentence in the introduction to be a bit more nuanced. We also modified the conclusion based on comment #1 and have added a sentence in the beginning of the second paragraph in the discussion highlighting the need to consider other factors beyond feeding and nutritional interventions (based on reviewer #1's comment #2).

Comment #4: Methods - Lines 50-52. In terms of infants excluded, is this data available for analysis/was there any follow up of these infants? There is a lack of outcome data on infants with disability and as a result a lack of visibility and guidance on their management. This was raised in recent WHO guideline development group meeting in Geneva and reflected in publication recently. Not related to this paper but as a bonus activity, I encourage the authors to analyse data of available or share with WHO (who are undertaking a pooled data analysis to stratify risk in infants and children).

Engl, M., et al. (2022). "Children living with disabilities are neglected in severe malnutrition protocols: a guideline review." Archives of Disease in Childhood 107(7): 637-643.

Response #4: Great to hear that WHO is undertaking this effort; thank you for sharing. We recognize the importance of collecting data on additional vulnerable groups including those with disabilities and severe illness. A recent webinar and set of resources released by USAID highlighted the same need and importance. In our study, data for infants with disabilities are not available beyond the eligibility questionnaire since excluded infants were not enrolled and followed up in this study, mainly because their conditions would interfere with feeding.

Comment #5: Methods - Lines 14-15: Stunted/stunting and wasting/wasted terms used interchangeably. Suggest to use wasting or stunting when referring to concept or process and wasted/stunted when referring to measures at timepoints. For example:

Current: Six-month stunting, underweight and wasting were defined by z-scores $<-2SD$ for length-for-age (LAZ), weight-for-age (WAZ) and weight-for-length (WLZ), respectively.

Possible revision: Six-month stunted, underweight and wasted were defined by z-scores $<-2SD$ for length-for-age (LAZ), weight-for-age (WAZ) and weight-for-length (WLZ), respectively.

Response #5: We have made the suggested change in this sentence and throughout. We hope that the change was implemented in the correct way throughout.

Comment #6: Methods - Lines 15-16: Defines SAM as WLZ <-3 . Is there a reason for using SAM terminology here rather than 'severe wasting' since this is the first time it referred to, and it is 'wasting' rather than 'SAM' that you explore in the analysis.

Response #6: You are correct; thank you for recognizing this. We have removed this sentence since we do not report on this outcome in the paper.

Comment #7: Results - Lines 14-18: Was bilateral oedema checked?

Response #7: We did check bilateral oedema in this study as a danger sign of malnutrition that would be a reason for referral as part of the safety net. A description of the safety net procedures has been added to the methods section.

Comment #8: Results - Line 32 What was correlation between reported feeding difficulties and growth outcomes?

Response #8: We have addressed this suggestion based on reviewer #1's comment #9. We conducted the suggested analyses and have added the results to a new supplementary table (S7) and the text in the results section after Figure 2. Please see Table S7 below for easy reference.

Table S7. Feeding difficulties in the first six months and poor growth outcomes at six months

	n (%) of poor growth outcomes among infants whose mothers ever reported feeding difficulties in first 6 months of the infant's life	Unadjusted			Adjusted*		
		N	RR (95% CI)	p-value	N	RR (95% CI)	p-value
Stunted at 6 months	153 (37.1%)	858	1.21 (0.99-1.48)	0.063	833	1.13 (0.92-1.39)	0.227
Underweight at 6 months	127 (30.6%)	860	1.55 (1.22-1.98)	<0.001	835	1.39 (1.09-1.78)	0.01
Wasted at 6 months	46 (11.1%)	859	1.49 (1.00-2.21)	0.05	834	1.36 (0.91-2.03)	0.134

*Adjusted by maternal education, maternal age, parity, place of residence, wealth quintile, birthcount, sex, LBW type, site; and with cluster-robust standard errors for clustering by mother.

Comment #9: Results - Lines 3-5: How did Term SGA v Preterm SGA compare in terms of risk of poor growth outcomes?

Response #9: Thank you for this important question; we have investigated it and share insights below. Before sharing the results, we wanted to mention that we chose preterm AGA as the reference group for the LBW type indicator because preterm AGA infants had the lowest risk for poor growth outcomes overall compared to the other LBW types; this was our consistent approach for defining reference groups for all indicators. When using preterm SGA as a reference in the analysis, we found the following: Term SGA infants had an increased risk for being stunted (RR: 1.23, 95% CI: 0.96, 1.58), underweight (RR: 1.25, 95% CI: 0.92, 1.70) and wasted (RR: 1.58, 95% CI: 0.83, 3.00) at six months respectively, but this risk was not statistically significant compared to preterm SGA infants. We also examined the risk for poor six-month growth outcomes among AGA and LGA infants compared to SGA infants and found that AGA infants had a statistically significant decrease in the risk for being stunted (RR: 0.46, 95% CI: 0.35, 0.62) and underweight (RR: 0.43, 95% CI: 0.29, 0.63) at 6 months compared to SGA infants. Finally, we explored the risk for poor six-month growth outcomes among infants born preterm versus term and found that preterm infants were at statistically significant risk for all three poor growth outcomes at six months compared to term infants [(stunted RR: 1.87, 95% CI: 1.51, 2.31); (underweight RR: 2.09, 95% CI: 1.64, 2.74); (wasted RR: 1.89, 95% CI: 1.20, 2.99)]. Overall, preterm AGA infants appear to be best off. All these explorations and findings highlight the need to pay attention to the LBW type, that takes into account timing of birth (preterm/term) and size for gestational age at birth (SGA/AGA/LGA), as a risk factor for poor growth outcomes rather than just preterm or just SGA alone. We have highlighted this finding in our paper already. We have elected not to add these new explorations and results to the paper as we believe it would add inconsistency and unnecessary complexity.

Comment #10: Results - Lines 40-41: Suggest as per comment on terminology above;

Current: Stunting, underweight, and wasting were more prevalent among infants failing to regain their

birthweight by two weeks versus those regaining.

Revised: Stunted, underweight, and wasted were more prevalent among infants failing to regain their birthweight by two weeks versus those regaining.

Response #10: Thank you for the suggestion; we have made the change.

Comment #11: Results - Lines 42 and 44: Suggest as per comment on terminology above.

Current: In multivariable models, infants not regaining birthweight by two weeks had 1.51 (1.23-1.85) and 1.55 (1.21-1.99) times greater risks of six-month stunting and underweight, respectively, compared to infants regaining.

Revised: In multivariable models, infants not regaining birthweight by two weeks had 1.51 (1.23-1.85) and 1.55 (1.21-1.99) times greater risks of six-month stunted and underweight, respectively, compared to infants regaining.

Response #11: Thank you for the suggestion; we have made the change.

Comment #12: Results - Lines 15-19: It would be valuable to know the association between reported feeding difficulties and six month poor growth outcomes.

Response #12: Please see our response to comment #8 above.

Comment #13: Lines 25-27:

Current: Preterm-SGA and term-SGA infants exhibited greater risks for six-month stunting and underweight compared to preterm-AGA infants.

Revised: Preterm-SGA and term-SGA infants exhibited greater risks for six-month stunted and underweight compared to preterm-AGA infants.

Response #13: Thank you for the suggestion; we have made the change.

Comment #14: Discussion - Lines 28-29: Suggest refine to include 'early' and to refer to 'small' as well as nutritionally at risk as some LBW infants may not initially be identified as wasted/stunted/underweight/growth faltering. Suggested revision to read:

These findings emphasize the need for early growth monitoring and proactive intervention for small and/or nutritionally at-risk infants.

Response #14: Thank you for the comment; we have made the change.

Comment #15: Discussion - Lines 29-30. It would also be valuable to discuss the association between reported feeding difficulties and poor growth outcomes at 6 months. Nearly half of all mothers reported feeding difficulties. The MAMI Care Pathway, referred to earlier, includes reported feeding difficulties as criteria for further assessment and enrollment in support.

Response #16: Please see our response to comment #8 above. We have also added the following sentence to the end of the first paragraph in the discussion: "Additionally, nearly half of mother-infant dyads reported feeding difficulties in the first six months that reinforces the need for universal and consistent lactation support targeted to the needs of LBW infants in low-resource settings. The MAMI Care Pathway Package highlights reported feeding difficulties as criteria for further assessment and enrollment in support (<https://www.enonline.net/attachments/4004/MAMI-Care-Pathway-Package-Document-07June2021.pdf>). We have also include a mention of the recently released WHO recommendations for care of the preterm or low-birth-weight infant.

Comment #16: Discussion - Lines 35-37: Note that risk stratification by gestational age is a feature of pooled data analysis underway by WHO that would be useful to refer to here. More details are available from Nigel Rollins, WHO.

Response #16: This is great to hear; thank you for sharing. Since the pooled data analysis underway by the WHO is yet to be published, we were not able to reference it. However, we have noted current work that is underway by the WHO and Lancet Vulnerable Infant group in the second paragraph of the discussion. We will also plan to connect with Dr. Nigel Rollins at the WHO separately.

Comment #17: Discussion - Lines 37-39: Improved GA dating, etc will only be available in limited LMICs. In reality, capacity to identify LBW, never mind sub--types, will be limited/non-existent. I think an additional reflection (and a key learning from this paper) could be that it is very important for those running prevention and/or treatment services for wasting, stunting and underweight recognise that they are dealing with at risk infants and children with differing birth histories and early growth trajectories that have (and will continue) to contribute to poor growth (and most importantly, poor functional) outcomes in both early and later life. There needs to be much more integrated guidance and continuity of care provision between reproductive/maternal/neonatal health services that identify higher risk infants and their mothers, and subsequent community based interventions and active responsive growth surveillance.

Response #17: We agree with your statement that GA dating may be challenging to implement and have noted this as a limitation in the discussion. However, GA is critical for public health interventions as noted in the MAMI framework and in the upcoming Lancet Series. There is a push to move beyond birthweight and we hope that this will be possible in the future. In the meantime, for settings that can employ GA dating, this information and further risk stratified interventions will be relevant. Thank you for your reflection and appreciate the framing. We very much agree. We have added text to reflect your suggestion in the second paragraph of the discussion section.

Comment #18: Discussion - Lines 4-8: These are important factors but centre on specific quite technical components of feeding and does not take into account the wider determinants/interactions on growth and requiring a more holistic/cross service interventions. Nearly half of mothers reported feeding difficulties. It would be important to research further what type of feeding support they got (the study reports in general terms quite a range from individual to group sessions). Other important (and highly relevant) factors will include clinical care, maternal nutrition and health including mental wellbeing, and the role of wider family and community support to mothers in their central role in young infant growth and development.

Response #18: These are all very important and relevant points. We have addressed some of these in previous responses to earlier comments and in the additions to the discussion. Additionally, we will take up some of this in the next steps in terms of either additional analysis of the data collected or as part of future studies that will build on the knowledge gained through the LIFE study. Lactation support is one such area of future research that we are embarking on. We also intend to write a separate manuscript on social support related to LBW infant feeding and care.

Comment #19: Discussion - Lines 25-26: An additional valuable reflection may be that exclusion of infants with congenital abnormalities, while justified for study purposes, does mean that the particular needs and likely poorer growth outcomes associated with early life disability is not captured in this analysis.

Response #19: This is absolutely true. Please see our response to your comment #4. We have added a sentence with your suggested text as a limitation in the discussion section.

Comment #20: Discussion - Lines 44 and 45: Early indicators are need but also close collaboration between early postnatal care and subsequent surveillance and follow up in the community for LBW infants.

Response #20: Thank you for making this important point and we completely agree. We have added text in the second paragraph of the discussion section to reflect this and a related comment above.

Comment #21: Discussion - Lines 44-46: Suggest to use phrase 'growth faltering' rather than growth failure, which is more consistent with the approach reflected in the paper and analysis.

Hoehn C, Lelijveld N, Mwangome M, Berkley JA, McGrath M, Kerac M. Anthropometric Criteria for Identifying Infants Under 6 Months of Age at Risk of Morbidity and Mortality: A Systematic Review. *Clin Med Insights Pediatr.* 2021 Oct 21;15:11795565211049904. doi: 10.1177/11795565211049904. PMID: 34707425; PMCID: PMC8543668.

Note also this paper:

Mwangome, M., et al. (2019). "Anthropometry at birth and at age of routine vaccination to predict mortality in the first year of life: A birth cohort study in BukinaFaso." *PLoS One* 14(3): e0213523.

Mwangome, M., et al. (2021). "Growth monitoring and mortality risk in low birthweight infants: a birth cohort study in Burkina Faso [version 1; peer review: 1 approved with reservations]." *Gates Open*

Research 5(82).

Response #21: Thank you for the suggestion and for sharing these papers; we have made the change in the discussion.

Comment #22: Conclusions - Lines 33-34: Regarding the following statement: "LBW type and early growth may be important parameters that could be used to identify and proactively manage nutritionally at-risk infants."

There is a risk that the emphasis on identification of LBW type is necessary to identify and manage at risk infants. Whilst ideal, this pursuit of the ideal may be the enemy of the good. This study shows that many of these particularly at risk LBW types are 'captured' by anthropometric indicators, such as underweight (which is a feasible, well established and evidenced indicator of risk of poor outcomes). For example, note systematic review that indicates weight for age, followed by MUAC, as indicators that identify infants under 6 months at risk of poor outcomes. For example:

Hoehn C, Lelijveld N, Mwangome M, Berkley JA, McGrath M, Kerac M. Anthropometric Criteria for Identifying Infants Under 6 Months of Age at Risk of Morbidity and Mortality: A Systematic Review. *Clin Med Insights Pediatr.* 2021 Oct 21;15:11795565211049904. doi: 10.1177/11795565211049904. PMID: 34707425; PMCID: PMC8543668.

Note also this paper:

Also:

Mwangome, M., et al. (2019). "Anthropometry at birth and at age of routine vaccination to predict mortality in the first year of life: A birth cohort study in BurkinaFaso." *PLoS One* 14(3): e0213523.

Given that, I think the conclusions should reflect this, perhaps noting that early identification of growth problems in infants under 6 months will identify higher risk infants that warrant close surveillance and prompt support. Infants who are known to be small at birth should be automatically enrolled in supportive care. Such an approach is reflected in the MAMI Care Pathway approach, referred to earlier.

Response #22: Thank you for your reflections and for sharing relevant papers. We agree that identification of LBW type is not the only strategy for identifying and managing at risk infants and recognize existing efforts such as MAMI. We have now cited the MAMI Care Pathway in a couple areas within the manuscript per your very accurate suggestions. We are suggesting two indicators that can be used to identify risk early and, where relevant, justify proactive intervention. The infants in our cohort are all small and at-risk, but given the resource constraints, we want to find ways to prioritize those who may require more attention and possibly intervention earlier.

Comment #23: Conclusions - Lines 34-38:

Regarding the following conclusions: "Additionally, utilization of these indicators could facilitate the prioritization of scarce resources (e.g., facility staff, space, and breast pumps) and services (e.g., lactation support) to infants at highest risk. Research is needed to support optimal feeding strategies for LBW infants and evaluate the role of human milk volume, nutrient composition, and feeding frequency on infant growth."

In practice, early growth patterns and anthropometric indicators and birth history are more likely to be feasible compared to identifying sub-types of LBW (not detracting from the value these would have where there is resource and capacity to measure these).

Throughout the paper there is no mention of the role of the mother's nutrition, health and wellbeing in determining growth outcomes of her infant. Some data on the mother is gathered, eg age, but there is no reflection on this. Appreciating the focus is on infant parameters and therefore conclusions are accordingly, this could perhaps be included in the limitations section.

Response #23: Thank you for this important comment. We have added relevant text to the second paragraph of the discussion which addresses this comment and similar comments above from yourself and the other reviewer.

Comment #24: Figure 1: Amongst infants excluded, quite a number were excluded as 6-month follow-up not feasible 745 (72%). What were the factors that made it not feasible?

Response #24: This is a great question. As part of the eligibility screening survey conducted within 72 hours of birth, mothers were asked (in addition to other eligibility criteria): "Is it feasible to complete follow-up until 6 months?" If they answered no, this was an automatic reason for exclusion. The main reason for answering no to this question was that many women (more prevalent in the Indian sites) delivered in a health facility near their mother's home but they return to their husband's family home after discharge, which may have been too far from the facility in which they gave birth. Unfortunately, we did not collect specific reasons for this for each dyad, which is a lesson learned for future eligibility screening surveys.

Comment #25: Tables: Note suggested revision to terminology (e.g. stunted rather than stunting at 6 months).

Response #25: We have changed "stunting" to "stunted" and "wasting" to "wasted" in the Tables.

VERSION 2 – REVIEW

REVIEWER	Abdallah, Yaser Makerere University, Paediatrics and Child Health
REVIEW RETURNED	05-Jan-2023
GENERAL COMMENTS	i believe the conclusion and recommendation for this study should be SGA infants whether term or preterm and infants who fail to regain birth weight by 2 weeks are at risk for growth faltering. Addressing early nutrition to enable birth weight recovery by 2 weeks is crucial. Further studies are needed to understand nutritional needs and growth patterns for SGA.

VERSION 2 – AUTHOR RESPONSE

Reviewer #1 – Miss Yaser Abdallah

Comment #1: I believe the conclusion and recommendation for this study should be SGA infants whether term or preterm and infants who fail to regain birthweight by 2 weeks are at risk for growth faltering. Addressing early nutrition to enable birth weight recovery by 2 weeks is crucial. Further studies are needed to understand nutritional needs and growth patterns for SGA.

Response #1: Thank you for taking the time to review our manuscript second time. We have made edits to the conclusion in the abstract and in the main text to reflect your comments. We specifically called out SGA infants when mentioning that LBW type may be an important parameter to identify at-risk infants. In that same sentence, we replaced "early growth" with "lack of birthweight regain by two weeks." We also specified that we are referring to the need for early identification and intervention. Finally, we updated the last sentence to address your suggestion regarding further research regarding nutritional needs and growth patterns of SGA infants.